# Increasing trends of pharmaceutical payments to breast cancer specialists in Japan: A retrospective study from 2016 to 2019

**Yudai Kaneda**[1]*, **Erika Yamashita**[2], **Hiroaki Saito**[3], **Kenji Gonda**[4], **Masahiro Wada**[5], **Tetsuya Tanimoto**[2,6], **Akihiko Ozaki**[2,4]

1 School of Medicine, Hokkaido University, Hokkaido, Japan, 2 Medical Governance Research Institute, Tokyo, Japan, 3 Department of Internal Medicine, Soma Central Hospital, Fukushima, Japan, 4 Breast and Thyroid Center, Jyoban Hospital of Tokiwa Foundation, Fukushima, Japan, 5 Department of Breast Surgery, Utsunomiya Central Clinic, Tochigi, Japan, 6 Department of Internal Medicine, Navitas Clinic Kawasaki, Kanagawa, Japan

* nature271828@gmail.com

## Abstract

### Introduction

The introduction of new drugs often leads to aggressive promotion and potential financial conflicts of interest, which may bias treatment decisions and potentially harm patients. The breast cancer therapeutics market is rapidly evolving globally, and Japan is no exception. This study aimed to analyze trends in pharmaceutical payments to breast cancer specialists in Japan from 2016 to 2019, focusing on company-level data, relationships with new drug introductions, and individual specialist payment patterns.

### Methods

This retrospective study examined financial relationships between pharmaceutical companies and breast cancer specialists in Japan from 2016 to 2019. The analysis focused on certified specialists as of May 2023 and used payment data from 93 pharmaceutical companies for activities such as lecturing, writing, and consulting. First, a company-level analysis examined total payments, categories, and trends for all companies and the top 10 individually; second, a specialist-level analysis looked at payment amounts amount and counts. The Gini index was employed to assess the concentration of payments among specialists.

### Results

Total payments reached USD 13,329,911, growing at 10.1% annually, with 81.4% allocated to lecturing engagements. The top 10 companies, led by Chugai Pharmaceutical, Eisai, and AstraZeneca, accounted for 89.5% of all payments. Companies like Pfizer Japan and Eli Lilly Japan saw notable increases following the introduction of new drugs such as palbociclib and abemaciclib. Payment distribution was highly skewed, with an average of $7,692 per

**Data Availability Statement:** All relevant data are within the manuscript and its Supporting Information files.

**Funding:** This study was funded in part by the Medical Governance Research Institute. This non-profit enterprise receives donations from a dispensing pharmacy, namely Ain Pharmacies, Inc.; other organizations; and private individuals. This study also received support from Tansa (formerly known as the Waseda Chronicle), an independent nonprofit news organization dedicated to investigative journalism. There was no grant number for the donation to the Medical Governance Research Institute.

**Competing interests:** Dr. Ozaki received personal fees from Medical Network Systems, Kyowa Kirin Inc., Becton, Dickinson and Company, Pfizer Inc, and Taiho Pharmaceutical Co., Ltd., outside the scope of the submitted work. Hiroaki Saito received personal fees from Taiho Pharmaceutical Co. Ltd. outside the scope of the submitted work. Tetsuya Tanimoto received personal fees from Medical Network Systems and Bionics Co. Ltd., outside the scope of the submitted work. All remaining authors have nothing to disclose. Regarding non-financial conflicts of interest among the study authors, Dr. Ozaki is engaged in ongoing research examining financial and non-financial conflicts of interest among healthcare professionals and pharmaceutical companies in Japan and other countries.

specialist but a median of only $2,884. A Gini index of 0.994 further confirmed that a small group of specialists received a disproportionately large share of the payments

## Conclusion

From 2016 to 2019, pharmaceutical payments to Japanese breast cancer specialists increased significantly, coinciding with new drug introductions. The concentration of payments among a select group of specialists raises concerns about potential influences on clinical decision-making and guideline recommendations.

## Introduction

The primary concern of healthcare professionals (HCPs) and healthcare organizations (HCOs) is the health and well-being of patients, and patient-centered care is a fundamental concept in contemporary medicine [1, 2]. However, financial relationships with pharmaceutical companies can pose significant financial conflicts of interest (FCOIs) for HCPs and HCOs [3–5], making the management of such relationships a crucial aspect of healthcare policy. In fact, financial ties between medical professionals and pharmaceutical companies have been reported to potentially influence physicians' prescribing patterns, clinical guideline recommendations, and clinical research unintentionally, necessitating proper FCOI management [6–11]. Rather than prohibiting interactions outright, promoting transparency is seen as a pragmatic approach, and many high-income countries have implemented transparency initiatives based on self-regulation or legal frameworks [12–17]. In Japan as well, to enhance transparency regarding FCOIs in healthcare, the Japan Pharmaceutical Manufacturers Association (JPMA), the largest pharmaceutical industry association in the country, has mandated since 2013 that all member companies disclose all payments to physicians, including for lectures, writing, and consulting, on their respective websites, listing individual names, affiliations, and payment amounts, similar to the Sunshine Act [18]. However, the current situation in Japan is such that there is no public disclosure of data indicating which specific payments are associated with particular pharmaceutical products, and furthermore, expenses related to accommodation and lodging are also not disclosed on an individual payment level.

In Japan, the breast cancer incidence has been rising, with 95,620 cases recorded in 2018, making it the most common cancer among Japanese women [19, 20]. Furthermore, Japan sees many recently introduced drugs, including treatments like olaparib (Lynparza®) available for patients with BRCA mutations, CDK4/6 inhibitors (abemaciclib [Verzenio®] and palbociclib [Ibrance®]) for hormone receptors positive diseases, and antibody-drug conjugates (trastuzumab deruxtecan [Enhertu®] and trastuzumab emtansine [Kadcyla®]) [21–23]. Therapies responsive to neoadjuvant chemotherapy, exemplified by CREATE-X and KATHERINE Trial, are becoming mainstream, especially for TNBC and HER2-positive breast cancer subtypes [24–26].

Given these factors, the breast oncology sector is expected to be a growing market for pharmaceutical companies, both globally and in Japan [27]. Indeed, the global breast cancer market is projected to reach approximately $73.68 billion by 2032, with a compound annual growth rate (CAGR) of 9.9% from 2023 [28]. Similarly in Japan, the industry share for breast cancer treatments is expected to grow at a CAGR of 9.6%, reaching over $2 billion by 2028 [29, 30]. This change aligns with the projected growth of Japan's mammography market (CAGR of

8.84%, reaching $193.09 million by 2028 [31]), reflecting the importance of early diagnosis and treatment through mammography in reducing breast cancer mortality by about 15% [32–34].

However, despite recent advancements in pharmaceuticals for breast cancer and many drugs showing significant results for surrogate markers, only a limited number have demonstrated statistically significant effects in extending survival rates. Indeed, most of the CDK and PARP inhibitors introduced in Japan have not been able to show an extension in survival rates as the pre-determined primary outcome [31]. In this context, pharmaceutical companies are expected to focus promotional payments on breast cancer specialists in Japan to gain market advantage and boost sales, mirroring trends in hematology and respiratory specialties [35, 36]. Such FCOIs can significantly influence the choice of treatment strategies for breast cancer patients.

For example, in the treatment of advanced triple-negative breast cancer expressing PD-L1, atezolizumab (Tecentriq®), and in cases of unresectable or recurrent breast cancer, bevacizumab (Avastin®), both received accelerated approval from the United States Food and Drug Administration; however, the efficacy of these drugs was not confirmed in subsequent confirmatory trials, and their approvals were withheld in the United States [37–39]. On the other hand, Japan lacks a system for re-evaluating drug efficacy post-approval, and consequently, the 2022 Breast Cancer Guidelines in Japan continue to recommend these treatments [40, 41]. Therefore, given that physicians' treatment decision makings are heavily influenced by the guidelines' recommendations [42], the potential influence of substantial financial relationships between the authors of the guidelines and Chugai Pharmaceutical, the manufacturer of these drugs, cannot be ignored [7, 43, 44]. However, in Japan, there is no research evaluating the scale and trends of payments from pharmaceutical companies to breast cancer specialists.

This study primarily aimed to analyze the trends and distribution of pharmaceutical payments to breast cancer specialists in Japan from 2016 to 2019, focusing on company-level data. We sought to evaluate relationships between new drug introductions and changes in payment patterns. Additionally, we examined these payments from an individual specialist perspective to provide a comprehensive overview.

## Methods

### Participants

This retrospective study considered all of the breast cancer specialists certified by the Japanese Breast Cancer Society (JBCS) as of May 2023. The name list of specialists from previous years was not publicly available from the JBCS. Therefore, we used the latest version of the name list at the JBCS website at the data extraction for this study. It is a primary and large professional society in breast cancer treatment in Japan, with 8,552 JBCS members as of 2023, including 7,147 physicians and 1,405 non-physicians. The JBCS contributes to training and certifying physicians specializing in breast cancer in Japan, with the latest name list of the specialists publicly disclosed on the Society's webpage. The JBCS also contributes to the development of clinical practice guidelines for breast cancer, promotes research and continuing medical education (CME) in breast cancer, and publishes an English-language academic journal, Breast Cancer, which is issued six times a year and available online.

### Data collection

To examine the trends in pharmaceutical payments prior to the COVID-19 pandemic, we analyzed data from 2016 to 2019 extracted from 93 pharmaceutical companies affiliated with the Japan Pharmaceutical Manufacturers Association (JPMA), a major trade organization of Japanese pharmaceutical companies. During the creation of the overall payment data database, two

individuals collected data from publicly disclosed corporate data. One person was responsible for checking for any omissions in data collection and performing data recollection. Subsequently, five individuals were divided into two teams to perform checks and verify that there were no mistakes through mutual verification. Finally, the completed data was cross-checked against the finalized database, which was then completed. As most companies released data for individual years prior to and including 2018, we meticulously processed each year's information to ensure comprehensive coverage.

From there, data extraction regarding breast cancer specialists at each step was conducted with utmost care by one person per step. This data collection method followed the approach of our previous studies [35, 36, 45]. In summary, data collection involved gathering names, affiliations, and other details of specialists from the JBCS webpage in November 2021, followed by extracting their payment data from the entire database. In May 2023, all data collected were organized and accessed for research purposes, and analyses were conducted.

## Data analysis

The analysis focused on payments related to lecturing, writing, and consulting, as per the JPMA transparency guidelines. These guidelines require member companies to disclose payments to healthcare professionals, including individual consent for disclosure in advance. The study did not include personal payments for meals, education, travel, and accommodations, as these were aggregated by each company.

Our analysis first focused on company-level data. We began by investigating the total annual payment amounts and their categories across all companies for the entire study period. We then examined the temporal trend of payment amounts, both for all companies combined and for the top 10 companies individually, over the four-year study period. Additionally, we visually inspected the relationships between new drug introductions and changes in annual payments for companies that introduced new breast cancer drugs during the study period.

With regards to the specialist-level analysis, we conducted descriptive analyses of payment amounts, number of payments, and number of companies making payments, focusing on the specialists. We utilized the Gini index to assess the concentration of payments among specialists. In the specialist-level analysis, we conducted two separate analyses: one encompassing all companies, and another limited to companies with complete data for all four years.

For temporal trend analysis, we calculated the year-over-year changes in company-wise payment amounts, payment amounts per specialist, and the number of specialists receiving specific amounts of payments over the four-year period. First, we calculated the rate of change between each fiscal year (2017 vs 2016, 2018 vs 2017, 2019 vs 2018). To calculate the overall average yearly change, we then aggregated the three year-over-year rates and divided their sum by 3. The entire formula is as follows:

$$Average\ yealy\ change = \frac{\left(\frac{2017value}{2016value} - 1\right) + \left(\frac{2018value}{2017value} - 1\right) + \left(\frac{2019value}{2018value} - 1\right)}{3} \times 100$$

For the purpose of the study, Japanese yen (¥) was converted into US dollars ($) using the 2019 average monthly exchange rate of ¥109.0 per $1. Analyses were conducted with Microsoft Excel 16.0 (Microsoft® Corp., Redmond, WA) and Python 3.9.10.

## Ethical approval

This study was approved by the Ethics Committee of the Medical Governance Research Institute (approval number: MG2018-04-20200605; approval date: June 5, 2020). Since this study involved a retrospective analysis of publicly available information from pharmaceutical

**Table 1. Overview of pharmaceutical payments to breast cancer specialists in Japan, 2016–2019.**

| Variables | Metrics |
|---|---|
| **Total payments** | |
| Payment amounts (USD) | 13,329,910 |
| Number of payments | 18,501 |
| Number of companies making payments | 64 |
| **Category of payments, USD (%)** | |
| Lecturing | 10,849,291 (81.4) |
| Consulting | 1,260,781 (9.5) |
| Writing | 506,342 (3.8) |
| Other | 715,043 (5.4) |

companies and the Society's webpage, the Ethics Committee waived the requirement for informed consent.

## Results

Table 1 shows the overview of the obtained data. The data spanned from 2016 to 2019, captured 18,501 transactions from 64 pharmaceutical companies, accumulating USD 13,329,910.57. 81.4% of these financial interactions were attributed to speaking engagements.

Fig 1 shows the overall payments trend during the study period, indicating a monotonic increase in the amounts.

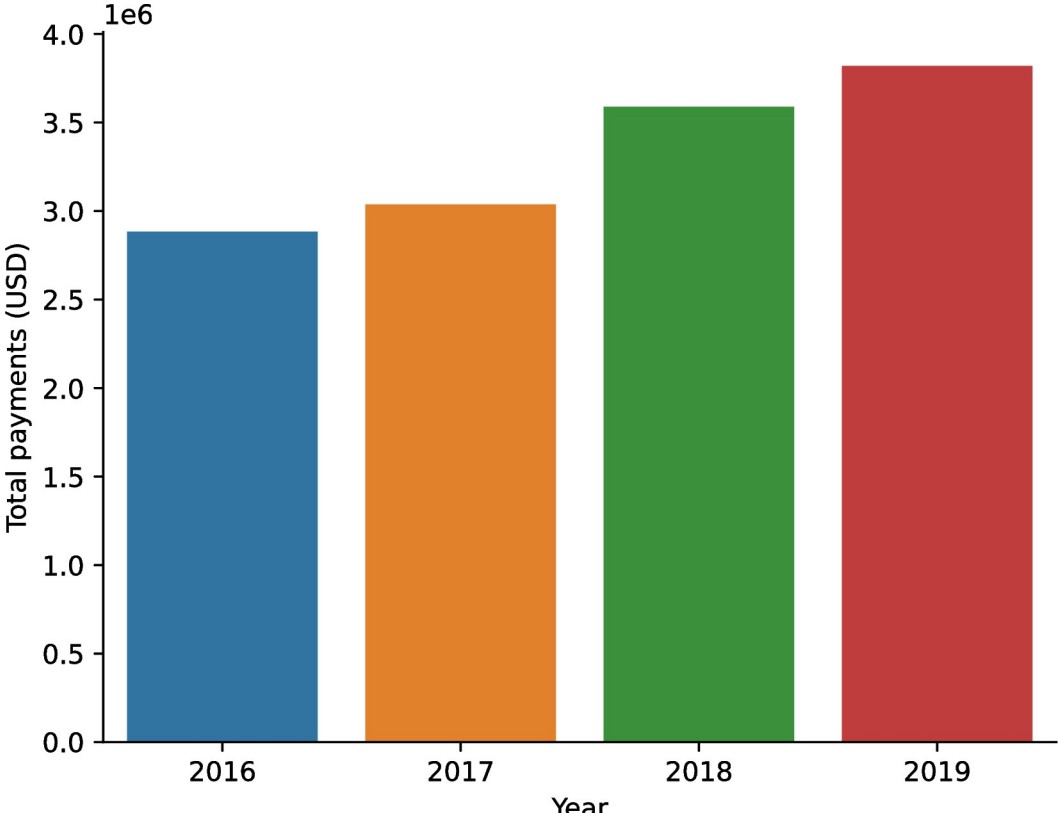

**Fig 1. Trend of total annual payments from pharmaceutical companies to breast cancer specialists in Japan, 2016–2019.**

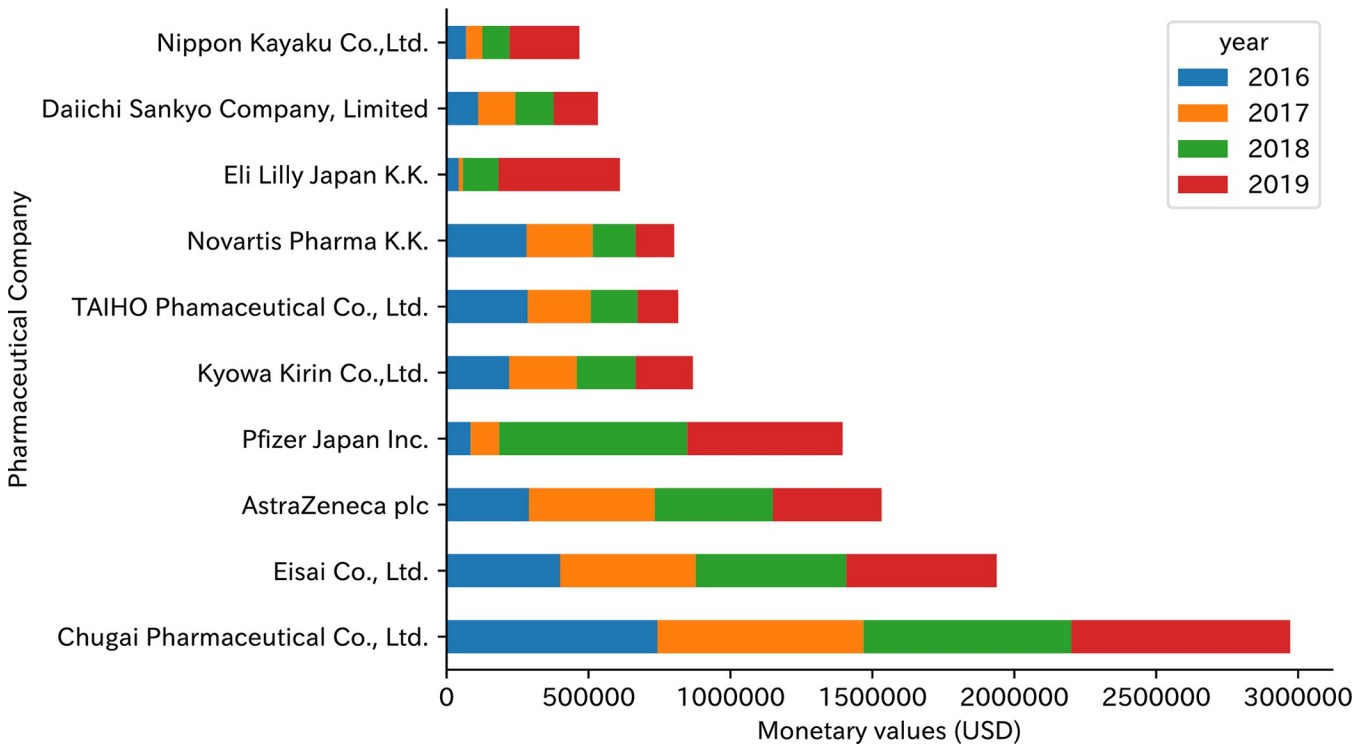

**Fig 2. Total payments and annual changes for top 10 pharmaceutical companies to breast cancer specialists in Japan, 2016–2019.**

Fig 2 displays the total and changes over four years for the top 10 companies with the largest monetary amounts. Among the 64 pharmaceutical companies providing payments to specialists, the contributions from the top 10 companies constituted 89.5% of the total payments, amounting to USD 11,932,018 between 2016 and 2019. Chugai Pharmaceutical led the list, followed by Eisai and AstraZeneca. Notably, Pfizer Japan experienced a notable increase in its payment amounts in the year 2018. Similarly, Eli Lilly Japan also augmented payments to specialists with a discernible rise in payments observed from the year 2018.

Fig 3 illustrates the relationship of the new drug introductions and changes in pharmaceutical company payments to breast cancer specialists in Japan during the study period. Palbociclib was introduced to the Japanese market in 2017, followed by an increase in Pfizer Japan's total payment amount in 2018. Similarly, abemaciclib was introduced to the Japanese market in 2018, followed by an increase in Eli Lily Japan's total payment amount in 2018 and 2019. There were no such tendencies among an introduction of olaparib and the payment amount of the AstraZeneca. Furthermore, it was difficult to investigate whether there was any increase in payments after the introduction of atezolizumab, as it was introduced in 2019.

Table 2 presents a comprehensive analysis of pharmaceutical payments to breast cancer specialists from 2016 to 2019. The average payment per specialist was USD 7,692, with a standard deviation of USD 24,162, while the median was USD 2,884 with an interquartile range (IQR) of USD 1,022–7,882, indicating a skewed distribution. The number of payments per specialist also varied considerably, with an average of 14 but a range from 1 to 303. The table shows that 75.4% of physicians received some form of payment. The proportion of physicians receiving payments decreased as the payment amount increased, with 68.6% receiving over USD 500, 27.0% over USD 5,000, and 1.6% receiving payments exceeding USD 100,000. The

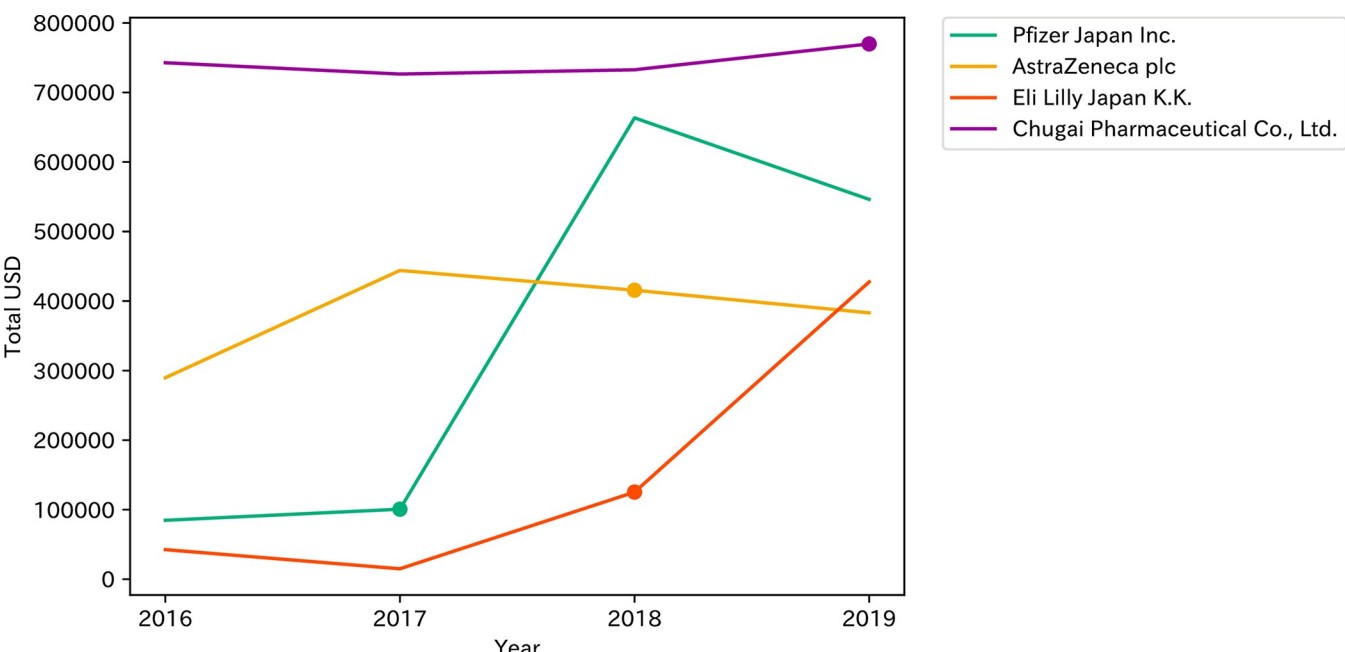

**Fig 3. New drug introductions and changes in pharmaceutical company payments to breast cancer specialists in Japan, 2016–2019.** Lines represent total annual payments, while dots indicate drug introduction dates. Palbociclib (Pfizer Japan) was introduced in 2017, abemaciclib (Eli Lilly Japan) and olaparib (AstraZeneca) in 2018, and atezolizumab (Chugai Pharmaceutical) in 2019).

**Table 2. Payment data to breast cancer specialists from pharmaceutical companies during the study period.**

| Variables | Metrics |
| --- | --- |
| **Per specialist averages (Mean ± SD)** | |
| Payment amounts (USD) | 7,692 ± 24,162 |
| Number of payments | 14 ± 26 |
| Number of companies making payments | 9.0 |
| **Per specialist medians (IQR)** | |
| Payment amounts (USD) | 2,884 (1,022–7,882) |
| Number of payments | 6 (2–14) |
| Number of companies making payments | 5 (2–12) |
| **Per specialist ranges** | |
| Payment amounts (USD) | 102–520,314 |
| Number of payments | 1–303 |
| Number of companies making payments | 1–90 |
| **Distribution of physicians by payment amount** | |
| Any payments, n (%) | 1,306 (75.4) |
| > USD 500, n (%) | 1,189 (68.6) |
| > USD 1000, n (%) | 1,001 (57.8) |
| > USD 5000, n (%) | 468 (27.0) |
| > USD 10000, n (%) | 266 (15.3) |
| > USD 50000, n (%) | 53 (3.1) |
| > USD 100000, n (%) | 27 (1.6) |
| **Gini index** | 0.994 |

SD = standard deviation; IQR = interquartile range

**Table 3. Trends of pharmaceutical payment breakdowns across four years.**

| Variables | 2016 | 2017 | 2018 | 2019 | Average yearly change (%) |
|---|---|---|---|---|---|
| **All pharmaceutical companies (N = 64)** | | | | | |
| **Total payments, USD (JPY)** | 2,883,945 (314,350,000) | 3,037,888 (331,129,838) | 3,589,583 (391,264,579) | 3,820,041 (416,384,464) | 10.1 |
| **Per specialist payment amount average (USD) (Mean ± SD)** | 3,169 ± 7,652 | 3,242 ± 7,346 | 3,881 ± 8,452 | 4,034 ± 9,292 | 8.6 |
| **Per Specialist Payment Amount Median (USD) (IQR)** | 1,136 (511–2,646) | 1,124 (525–2,802) | 1,334 (525–3,340) | 1,237 (536–3,195) | - |
| **Per Specialist Payment Amount Range, USD** | 102–134,164 | 102–126,022 | 92–129,453 | 102–130,675 | - |
| **Gini index** | 0.993 | 0.993 | 0.994 | 0.993 | - |
| **Distribution of physicians by payment amount** | | | | | |
| Any payments, n (%) | 910 (52.5) | 937 (54.1) | 925 (53.4) | 947 (54.6) | 1.4 |
| > USD 500, n (%) | 752 (43.4) | 783 (45.2) | 788 (45.5) | 844 (48.7) | 4.0 |
| > USD 1000, n (%) | 518 (29.9) | 531 (30.6) | 566 (32.7) | 572 (33.0) | 3.4 |
| > USD 5000, n (%) | 123 (7.1) | 120 (6.9) | 161 (9.3) | 149 (8.6) | 8.1 |
| > USD 10000, n (%) | 56 (3.2) | 63 (3.6) | 80 (4.6) | 83 (4.8) | 14.4 |
| > USD 50000, n (%) | 5 (0.3) | 3 (0.2) | 6 (0.3) | 5 (0.3) | 14.4 |
| > USD 100000, n (%) | 1 (0.1) | 1 (0.1)) | 1 (0.1) | 1 (0.1) | 0 |
| **Pharmaceutical companies with 4-year payment data (N = 34)** | | | | | |
| **Total payments, USD (JPY)** | 2,857,414 (311,458,105) | 3,016,587 (328,807,944) | 3,558,368 (387,862,074) | 3,798,523 (414,038,985) | 10.1 |
| **Per specialist payment amount average (USD) (Mean ± SD)** | 3,175 ± 7,678 | 3,240 ± 7,352 | 3,868 ± 8,440 | 4,037 ± 9,301 | 8.6 |
| **Per Specialist Payment Amount Median (USD) (IQR)** | 1,150 (511–2,624) | 1,095 (525–2,808) | 1,336 (525–3,319) | 1,239 (536–3,202) | - |
| **Per Specialist Payment Amount Range, USD** | 102–134,164 | 102–126,022 | 92–129,453 | 102–130,675 | - |
| **Gini index** | 0.992 | 0.993 | 0.994 | 0.993 | - |
| **Distribution of physicians by payment amount** | | | | | |
| Any payments, n (%) | 900 (51.9) | 931 (53.7) | 920 (53.1) | 941 (54.3) | 1.5 |
| > USD 500, n (%) | 747 (43.1) | 780 (45.0) | 785 (45.3) | 839 (48.4) | 4.0 |
| > USD 1000, n (%) | 517 (29.8) | 526 (30.4) | 561 (32.4) | 567 (32.7) | 3.2 |
| > USD 5000, n (%) | 121 (7.0) | 120 (6.9) | 160 (9.2) | 148 (8.5) | 8.3 |
| > USD 10000, n (%) | 56 (3.2) | 62 (3.6) | 79 (4.6) | 82 (4.7) | 14 |
| > USD 50000, n (%) | 5 (0.3) | 3 (0.2) | 6 (0.3) | 5 (0.3) | 14.4 |
| > USD 100000, n (%) | 1 (0.1) | 1 (0.1) | 1 (0.1) | 1 (0.1) | 0 |

Gini index for these payments was calculated at 0.994, indicating a high degree of concentration in the distribution of payments among the specialists.

Table 3 outlines the payment data to breast cancer specialists from pharmaceutical companies across four years. As visualized in Fig 1, total payments increased steadily from USD 2,857,414 (JPY 311,458,105) in 2016 to USD 3,016,587 (JPY 328,807,944) in 2017, then to USD 3,558,368 (JPY 387,862,074) in 2018, and finally to USD 3,798,523 (JPY 414,038,985) in 2019. This consistent growth resulted in an average annual increase of 10.1% over the four-year period.

In 2016, the median payment per specialist was USD 1,136 with an IQR of USD 511–2,646. By 2019, this median increased to USD 1,237 with an IQR of USD 536–3,195. This shift represents an annual increase rate of 8.6% in payments per specialist and a 1.4% increase in the number of specialists receiving payments.

The analysis thus focused on the remaining 34 companies with complete 4-year data, which showed significant annual increases of 8.6% in per specialist payments and 1.5% in the number of specialists with payments. Overall, the trends observed in the data from companies with complete four-year records were consistent with those seen in the data from all companies.

## Discussion

The increasing financial relationships between pharmaceutical companies and breast cancer specialists in Japan, as highlighted in this study, represent a significant trend in the pharmaceutical payments coming into the breast cancer care in Japan. The overall rise in payments, with a notable average annual growth of 10.1%, reflects the pharmaceutical industry's strategic efforts to engage healthcare professionals more deeply, as there has been a downward trend in HCP contact in the pharmaceutical industry in recent years, with HCPs becoming more selective in their interactions with pharmaceutical companies [46].

This increase is particularly pronounced in companies such as Pfizer Japan and Eli Lily Japan, which have expanded their payment allocations substantially. Specifically, Pfizer Japan's payments surged following the September 2017 introduction of palbociclib [47], and Eli Lilly Japan's payments increased significantly after introducing abemaciclib in November 2018 [48]. These increments, captured in the total payment growth from USD 2,857,414 in 2016 to USD 3,798,523 in 2019, underscore the competitive dynamics within the oncology sector, particularly in the domain of CDK4/6 inhibitors. Despite the similar therapeutic applications of these drugs to existing treatments [49], their aggressive promotion can be attributed to the marginal but clinically significant benefits they offer over competitors, specifically in terms of extension in progression-free survival rather than overall survival rates, a factor crucial in a highly competitive oncology market [50–52].

On the other hand, in the context of Chugai Pharmaceutical's financial engagements, their consistent high payments can largely be attributed to their portfolio of established therapies, particularly for HER2-targeting drugs such as trastuzumab (Herceptin®), pertuzumab (Perjeta®), and trastuzumab emtansine, as well as bevacizumab for HER2-negative cases. This may be the primary reason why a launch of atezolizumab did not increase the payments in 2019, although it is difficult to definitively investigate this within the study period of 2016 to 2019 [53]. However, the evolving competitive landscape, with new entrants like trastuzumab deruxtecan manufactured by Daiichi Sankyo, will pose a potential challenge to established players like Chugai Pharmaceutical. Trastuzumab deruxtecan has been approved not only for HER2-enriched breast cancer but also for low-HER2 breast cancer, positioning it as a competitor in both segments [54, 55]. Thus, there is a possibility that Chugai Pharmaceutical would increase its payments to the HCPs to protect its sales and share against Daiichi Sankyo. Similarly, Pfizer Japan's BRCA-targeted agent talazoparib (Talzenna®), which has been approved in Japan in January 2024, does not have a clear advantage over AstraZeneca's orapalib and may prompt increased investments by AstraZeneca [56].

The already escalating promotional payments from the pharmaceutical sector, which are likely to increase further, are contributing to further complexities in market dynamics. This trend underscores the need for a comprehensive analysis of promotional strategies and their implications for treatment decisions and guideline recommendations. Intriguingly, the most recent 2022 guidelines continue to endorse drugs such as atezolizumab and bevacizumab despite their withdrawal from the market in the United States [41]. Of note, the guideline was funded by the JBCS itself, and it has been reported that 115 out of 149 authors (77.2%) received at least one personal payment from pharmaceutical companies [57].

Furthermore, this study revealed a concerning trend in promotional payments in the field of breast oncology, akin to or potentially more pronounced than in other areas such as dermatology and respiratory medicine [35, 36], where payments are disproportionately concentrated among a select group of specialists. This skewed distribution raises the possibility that the recommendations in clinical guidelines might be influenced by authors who maintain robust relationships with pharmaceutical companies [11, 58]. Unfortunately, in the current Japanese context, individuals who receive very high payments from the pharmaceutical industry often hold prominent positions in medical societies [59]. This can result in biased guideline recommendations for breast cancer, thereby potentially affecting the objectivity of those guidelines; therefore, as Wright et al. have pointed out, payments exceeding $100,000 require careful scrutiny, even if they only constitute a small percentage [60].

Of note, pharmaceutical companies' promotional strategies heavily favor lecture-based engagements. This method enables the wide dissemination of information through a singular, concerted effort, providing a pragmatic approach for the involved companies. Indeed, such approaches have become vital for interaction and information acquisition among regional HCPs. However, reliance on these strategies raises substantial concerns regarding their broader impact on the medical community.

Furthermore, an important point to consider is the potential contribution of these strategies to the male-dominated nature of breast surgery field in Japan. The last author of this paper, who works in the field of breast surgery, has observed how gaining favor with pharmaceutical companies and demonstrating presence at their sponsored lectures play a significant role in career advancement within medical societies in Japan. However, the current scheduling of post-clinical evening lectures significantly hinders the participation of female physicians [61]. This scheduling conflict, combined with the dual demands of professional obligations and familial roles, often prevents Japanese women doctors from attending such events, thereby exacerbating gender disparities within the medical community. As a result, unlike other male-dominated surgical fields, although women make up about 40% of breast surgery specialists, men predominantly occupy responsible positions such as professors, directors, and presidents within medical societies. To rectify this situation, it is essential to explore more inclusive and accessible approaches to organizing educational events, ensuring equitable participation across genders, and accommodating the diverse needs of the medical community without reliance on pharmaceutical companies.

In addition, transparency has not significantly changed the behaviors of pharmaceutical companies and physicians; thus, it is important to recognize that while transparency is necessary, it alone is insufficient to address the negative impacts of pharmaceutical payments [62]. There is a report suggesting that the Open Payments system in the United States has even reduced public trust [63]. Therefore, completely eliminating FCOIs with pharmaceutical companies is challenging, necessitating exploring additional measures beyond transparency. Specifically, it would be an effective option for governing bodies, such as the Ministry of Education, Culture, Sports, Science and Technology, which oversees university hospitals, to intervene and set caps on payments to key figures in professional societies and guideline committees. Establishing standards at each university and hospital may also be effective.

This study has several limitations. First, the payment data was manually collected from 93 pharmaceutical companies, which introduces the possibility of errors in data gathering and reporting. Additionally, data extraction at each step was conducted by a single individual. Although each party involved exercised utmost caution to reduce mistakes and consistently shared information with the team, the possibility of human error could not be entirely avoided. Second, our focus was on direct payments for lecturing, writing, and consulting, excluding indirect financial relationships like meals, travel, and research grants, potentially

underrepresenting the total financial engagements. Additionally, the study used the most current list of breast cancer specialists without historical data for previous years, which might affect the accuracy of trend analyses. Also, the lack of detailed information on the purpose of payments and the specific drugs involved limits the depth of our analysis. Finally, the absence of a strict monitoring system for the accuracy of disclosed payments by pharmaceutical companies may question the reliability of the data, given the lack of penalties for non-disclosure under JPMA guidelines.

## Conclusion

The period from 2016 to 2019 marked a significant increase in payments from pharmaceutical companies to breast cancer specialists in Japan, reflecting both the introduction of new therapeutic agents and the ongoing value placed on established treatments. This study's findings, particularly the notable increases attributed to companies like Pfizer Japan and Eli Lilly Japan following the introduction of their CDK4/6 inhibitors, underscore the competitive dynamics within the oncology sector. The concentration of payments among a small group of specialists, as indicated by the Gini index, and the overall rise in individual payments raise important considerations regarding the influence of pharmaceutical funding on clinical decision-making.

As the breast oncology field continues to evolve, with new treatments and therapeutic strategies emerging, the management of financial relationships between healthcare professionals and the pharmaceutical industry remains a critical issue. Ensuring transparency and equity in these relationships is essential to maintain trust and integrity in the healthcare system, ultimately ensuring that patient care remains the foremost priority.

## Supporting information

**S1 Data.**
(XLSX)

## Acknowledgments

We express our gratitude to Mr Kohki Yamada for his constructive help on our analysis.

## Author Contributions

**Conceptualization:** Yudai Kaneda, Masahiro Wada, Akihiko Ozaki.

**Data curation:** Yudai Kaneda, Erika Yamashita, Akihiko Ozaki.

**Formal analysis:** Yudai Kaneda, Erika Yamashita, Akihiko Ozaki.

**Funding acquisition:** Akihiko Ozaki.

**Project administration:** Akihiko Ozaki.

**Writing – original draft:** Yudai Kaneda.

**Writing – review & editing:** Erika Yamashita, Hiroaki Saito, Kenji Gonda, Masahiro Wada, Tetsuya Tanimoto, Akihiko Ozaki.

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
