## [Decision Letter · Decision Letter 0]

23 Apr 2024

PONE-D-24-07542Increasing Trends of Pharmaceutical Payments to Breast Cancer Specialists in Japan: A Retrospective Study from 2016 to 2019PLOS ONE

Dear Dr. Kaneda,

Thank you for submitting your manuscript to PLOS ONE. After careful consideration, we feel that it has merit but does not fully meet PLOS ONE’s publication criteria as it currently stands. Therefore, we invite you to submit a revised version of the manuscript that addresses the points raised during the review process.

 The reviewer feedback is provided below.  Please submit your revised manuscript by Jun 07 2024 11:59PM. If you will need more time than this to complete your revisions, please reply to this message or contact the journal office at plosone@plos.org. Please include the following items when submitting your revised manuscript:A rebuttal letter that responds to each point raised by the academic editor and reviewer(s). You should upload this letter as a separate file labeled 'Response to Reviewers'.A marked-up copy of your manuscript that highlights changes made to the original version. You should upload this as a separate file labeled 'Revised Manuscript with Track Changes'.An unmarked version of your revised paper without tracked changes. You should upload this as a separate file labeled 'Manuscript'.If applicable, we recommend that you deposit your laboratory protocols in protocols.io to enhance the reproducibility of your results. Protocols.io assigns your protocol its own identifier (DOI) so that it can be cited independently in the future. For instructions see: https://journals.plos.org/plosone/s/submission-guidelines#loc-laboratory-protocols. Additionally, PLOS ONE offers an option for publishing peer-reviewed Lab Protocol articles, which describe protocols hosted on protocols.io. Read more information on sharing protocols at https://plos.org/protocols?utm_medium=editorial-email&utm_source=authorletters&utm_campaign=protocols.

We look forward to receiving your revised manuscript.

Kind regards,

Narcyz Ghinea

Academic Editor

PLOS ONE

Journal Requirements:

2. Thank you for submitting the above manuscript to PLOS ONE. During our internal evaluation of the manuscript, we found significant text overlap between your submission and previous work in the [introduction, conclusion, etc.].

Please revise the manuscript to rephrase the duplicated text, cite your sources, and provide details as to how the current manuscript advances on previous work. Please note that further consideration is dependent on the submission of a manuscript that addresses these concerns about the overlap in text with published work.

[If the overlap is with the authors’ own works: Moreover, upon submission, authors must confirm that the manuscript, or any related manuscript, is not currently under consideration or accepted elsewhere. If related work has been submitted to PLOS ONE or elsewhere, authors must include a copy with the submitted article. Reviewers will be asked to comment on the overlap between related submissions (http://journals.plos.org/plosone/s/submission-guidelines#loc-related-manuscripts).]

We will carefully review your manuscript upon resubmission and further consideration of the manuscript is dependent on the text overlap being addressed in full. Please ensure that your revision is thorough as failure to address the concerns to our satisfaction may result in your submission not being considered further.

"This study was funded in part by the Medical Governance Research Institute. This non-profit enterprise receives donations from a dispensing pharmacy, namely Ain Pharmacies, Inc.; other organizations; and private individuals. This study also received support from Tansa (formerly known as the Waseda Chronicle), an independent nonprofit news organization dedicated to investigative journalism. There was no grant number for the donation to the Medical Governance Research Institute."

"Dr. Ozaki received personal fees from Medical Network Systems, Kyowa Kirin Inc., Becton, Dickinson and Company, Pfizer Inc, and Taiho Pharmaceutical Co., Ltd., outside the scope of the submitted work. Hiroaki Saito received personal fees from Taiho Pharmaceutical Co. Ltd. outside the scope of the submitted work. Tetsuya Tanimoto received personal fees from Medical Network Systems and Bionics Co. Ltd., outside the scope of the submitted work. All remaining authors have nothing to disclose. 

Regarding non-financial conflicts of interest among the study authors, Dr. Ozaki is engaged in ongoing research examining financial and non-financial conflicts of interest among healthcare professionals and pharmaceutical companies in Japan and other countries."

5. We note that your Data Availability Statement is currently as follows: [All relevant data are within the manuscript and its Supporting Information files.]

Reviewers' comments:

Reviewer's Responses to Questions

**Comments to the Author**

1. Is the manuscript technically sound, and do the data support the conclusions?

Reviewer #1: Yes

Reviewer #2: Yes

2. Has the statistical analysis been performed appropriately and rigorously? 

Reviewer #1: Yes

Reviewer #2: Yes

3. Have the authors made all data underlying the findings in their manuscript fully available?

Reviewer #1: Yes

Reviewer #2: Yes

4. Is the manuscript presented in an intelligible fashion and written in standard English?

Reviewer #1: Yes

Reviewer #2: Yes

5. Review Comments to the Author

Reviewer #1: This manuscript looks at trends in the amount of payments to breast cancer specialists by pharmaceutical companies. It is an addition to the growing literature documenting payments from pharmaceutical companies to physicians working in a variety of specialties. Studies mainly, but not exclusively, out of the United States also show an association between the receipt of payments and prescribing behaviour. But transparency has not led to any substantial changes in the behaviour of either pharmaceutical companies or physicians. Therefore, while transparency is necessary it is not sufficient to deal with the negative effects of industry payments. In this light, it is necessary for the authors to propose policy measures to deal with the effects of the payments rather than just call for more transparency.

Line 99: In line with the comment above, the authors need to cite evidence that promoting transparency in isolation from other measures will achieve the desired outcome.

Line 104: In the US the disclosures have to name the product associated with the payments. Is that the case in Japan?

Lines 118-119: The authors need to cite literature showing that the recent introduction of medicines has resulted in a clinically meaningful increase in the length of survival and/or a clinically meaningful increase in the quality of life.

Line 152: How many people gathered the data. If it was more than one was it done independently and how were discrepancies resolved? If it was a single person how were errors avoided/detected?

Line 175: What statistical program, and its version number, was used?

Line 272: Neither reference 41 nor 42 discusses concrete evidence that newer CDK4/6 inhibitors offer clinically meaningful increases in overall survival time or quality of life.

Lines 293-294: Who financed the 2022 guidelines and how many of the people on the committee had COI with one or more makers of drugs used in the treatment of breast cancer?

Lines 310-320: Direct or indirect gender discrimination should not be tolerated, but the authors frame their argument around the assumption that attending post-clinical evening lectures sponsored by pharmaceutical companies will improve clinical care whereas the content of these talks is often biased in favour of the products made by the company sponsoring the talk.

Reviewer #2: 

Introduction:

- The author should have included a discussion about the potential conflict of interest that may arise from the relationship between the industry and oncologists, and how this can impact patient care. There is a substantial amount of literature on this topic. I understand the authors want to examine the trend of pharma payment, however, I am sure there already a relationship between Japanese breast oncologists and pharmaceutical companies hence mentioning how this can be detrimental to patient care in case not well controlled is important

- It would have been beneficial to note that these payments or conflicts of interest (COI) could potentially influence the prescribing of breast cancer medications with minimal or negligible efficacy. An example of this is Bevacizumab, which was subsequently withdrawn. Additionally, there is available data on this topic.

Results:

- Authors should consistently use commas after numbers in the thousands.

Discussion:

- The impact of the relationship between the industry and oncologists on patient care is not mentioned in the discussion.

- It could also improve the paper by mentioning the importance of transparency of pharma payments.

-The payment of >100,000 even though a small percentange needs to highlighted. There is a paper by Wright et al, JOP

6. PLOS authors have the option to publish the peer review history of their article (what does this mean?). If published, this will include your full peer review and any attached files.

Reviewer #1: **Yes: **Joel Lexchin

Reviewer #2: No

---

## [Author Response · Author response to Decision Letter 0]

30 Jun 2024

Narcyz Ghinea

Academic Editor

PLOS ONE 

May 9, 2024

ID: PONE-D-24-07542

Dear Editor,

 On behalf of my co-authors, I would like to express our gratitude to you and interest in our manuscript entitled “Increasing Trends of Pharmaceutical Payments to Breast Cancer Specialists in Japan: A Retrospective Study from 2016 to 2019” and for the opportunity to submit a revised version. 

 We hope you will find the revised version suitable for publication in PLOS ONE. We look forward to hearing from you shortly.

Sincerely,

Yudai Kaneda

School of Medicine, Hokkaido University

Nishi-7, Kita-15, Kita-ku, Sapporo, Hokkaido, Japan

Phone: +81-70-2231-7011

E-mail: nature271828@gmail.com

Increasing Trends of Pharmaceutical Payments to Breast Cancer Specialists in Japan: A Retrospective Study from 2016 to 2019

We would like to thank the Editors and Reviewers for their time and careful consideration of our manuscript. Please find below a detailed description of the revisions and our responses to the editors and reviewers.

Response to Reviewer: 1

1) This manuscript looks at trends in the amount of payments to breast cancer specialists by pharmaceutical companies. It is an addition to the growing literature documenting payments from pharmaceutical companies to physicians working in a variety of specialties. Studies mainly, but not exclusively, out of the United States also show an association between the receipt of payments and prescribing behaviour. But transparency has not led to any substantial changes in the behaviour of either pharmaceutical companies or physicians. Therefore, while transparency is necessary it is not sufficient to deal with the negative effects of industry payments. In this light, it is necessary for the authors to propose policy measures to deal with the effects of the payments rather than just call for more transparency.

Reply

Thank you for your insights. We agree that transparency alone does not substantially change the behaviors of pharmaceutical companies and physicians. As suggested, our manuscript now includes proposed policy measures, such as setting payment caps and establishing standards at institutional levels, to address the negative effects of industry payments more effectively than transparency alone.

Page 24 Line 367

“In addition, transparency has not significantly changed the behaviors of pharmaceutical companies and physicians; thus, it is important to recognize that while transparency is necessary, it alone is insufficient to address the negative impacts of pharmaceutical payments (62). There is a report suggesting that the Open Payments system in the United States has even reduced public trust (63). Therefore, completely eliminating COI with pharmaceutical companies is challenging, necessitating exploring additional measures beyond transparency. Specifically, it would be an effective option for governing bodies, such as the Ministry of Education, Culture, Sports, Science and Technology, which oversees university hospitals, to intervene and set caps on payments to key figures in professional societies and guideline committees. Establishing standards at each university and hospital may also be effective.”

References

62. Ozaki A, Murayama A, Saito H, Sawano T, Harada K, Senoo Y, et al. Transparency Is Not Enough: How Can We Improve the Management of Financial Conflicts of Interest Between Pharma and Healthcare Sectors? Clin Pharmacol Ther. 2021;110(2):289-91.

63. Sawano T, Ozaki A, Saito H, Shimada Y, Tanimoto T. Payments From Pharmaceutical Companies to Authors Involved in the Valsartan Scandal in Japan. JAMA Network Open. 2019;2(5):e193817-e.

2) Line 99: In line with the comment above, the authors need to cite evidence that promoting transparency in isolation from other measures will achieve the desired outcome.

Reply

Thank you for your comment. We acknowledge that transparency alone is insufficient and have revised our manuscript to include evidence supporting the need for additional measures alongside transparency. This revision emphasizes the limited impact of transparency alone and outlines the necessity of implementing further strategies, such as setting payment caps and establishing standards, to effectively manage conflicts of interest.

Page 24 Line 367

“In addition, transparency has not significantly changed the behaviors of pharmaceutical companies and physicians; thus, it is important to recognize that while transparency is necessary, it alone is insufficient to address the negative impacts of pharmaceutical payments (62). There is a report suggesting that the Open Payments system in the United States has even reduced public trust (63). Therefore, completely eliminating COI with pharmaceutical companies is challenging, necessitating exploring additional measures beyond transparency. Specifically, it would be an effective option for governing bodies, such as the Ministry of Education, Culture, Sports, Science and Technology, which oversees university hospitals, to intervene and set caps on payments to key figures in professional societies and guideline committees. Establishing standards at each university and hospital may also be effective.”

References

62. Ozaki A, Murayama A, Saito H, Sawano T, Harada K, Senoo Y, et al. Transparency Is Not Enough: How Can We Improve the Management of Financial Conflicts of Interest Between Pharma and Healthcare Sectors? Clin Pharmacol Ther. 2021;110(2):289-91.

63. Sawano T, Ozaki A, Saito H, Shimada Y, Tanimoto T. Payments From Pharmaceutical Companies to Authors Involved in the Valsartan Scandal in Japan. JAMA Network Open. 2019;2(5):e193817-e.

3) Line 104: In the US the disclosures have to name the product associated with the payments. Is that the case in Japan?

Reply

Thank you for your inquiry. In Japan, there is no public disclosure of data specifying which payments are associated with particular pharmaceutical products. We have also noted this in the text.

Page 8 Line 107

“However, the current situation in Japan is such that there is no public disclosure of data indicating which specific payments are associated with particular pharmaceutical products, and furthermore, expenses related to accommodation and lodging are also not disclosed on an individual payment level.”

4) Lines 118-119: The authors need to cite literature showing that the recent introduction of medicines has resulted in a clinically meaningful increase in the length of survival and/or a clinically meaningful increase in the quality of life.

Reply

Thank you for your comments. Despite recent advancements in pharmaceuticals for breast cancer, only a limited number of drugs have shown statistically significant effects in extending survival rates. For instance, most of the two CDK inhibitors and PARP inhibitors introduced in Japan have not demonstrated the ability to extend overall survival as the pre-determined outcome, despite positive results in many surrogate markers (27). We will clarify these points and cite the relevant literature in the revised manuscript.

Page 8 Line 120

“Of note, despite recent advancements in pharmaceuticals for breast cancer, and many surrogate markers have shown significant results, only a limited number of drugs have demonstrated statistically significant effects in extending survival rates. Indeed, most of the CDK and PARP inhibitors introduced in Japan have not been able to show an extension in survival rates as the pre-determined primary outcome (27).”

Reference

27. Goetz MP, Toi M, Huober J, Sohn J, Tredan O, Park IH, et al. Abemaciclib plus a nonsteroidal aromatase inhibitor as initial therapy for HR+, HER2- advanced breast cancer: Final overall survival results of MONARCH 3. Ann Oncol. 2024.

5) Line 152: How many people gathered the data. If it was more than one was it done independently and how were discrepancies resolved? If it was a single person how were errors avoided/detected?

Reply

Thank you for your inquiry regarding data collection. During the creation of the overall payment data database, two individuals collected data from publicly disclosed corporate data. One person was responsible for checking for any omissions in data collection and performing data recollection. Subsequently, five individuals were divided into two teams to perform checks and verify that there were no mistakes through mutual verification. Finally, the completed data was cross-checked against the finalized database. Then, data extraction at each step was handled by a single individual, and we acknowledge the potential for human error in this structure. Although each person involved exercised utmost caution to reduce mistakes and consistently shared information with the team, the possibility of human error could not be entirely avoided. We recognize this as a limitation of our study and have updated the limitations section accordingly to reflect this.

Page 12 Line 179

“During the creation of the overall payment data database, two individuals collected data from publicly disclosed corporate data. One person was responsible for checking for any omissions in data collection and performing data recollection. Subsequently, five individuals were divided into two teams to perform checks and verify that there were no mistakes through mutual verification. Finally, the completed data was cross-checked against the finalized database, which was then completed.”

Page 25 Line 381

“Additionally, data extraction at each step was conducted by a single individual. Although each party involved exercised utmost caution to reduce mistakes and consistently shared information with the team, the possibility of human error could not be entirely avoided.”

6) Line 175: What statistical program, and its version number, was used?

Reply

Thank you for your inquiry. The analyses were conducted using Microsoft Excel version 16.0 (Microsoft® Corp., Redmond, WA) and Python version 3.9.10.

Page 13 Line 213

“Analyses were conducted with Microsoft Excel 16.0 (Microsoft® Corp., Redmond, WA) and Python 3.9.10.”

7) Line 272: Neither reference 41 nor 42 discusses concrete evidence that newer CDK4/6 inhibitors offer clinically meaningful increases in overall survival time or quality of life.

Reply

Thank you for your comments. As you pointed out, the CDK4/6 inhibitors available in Japan have only slightly demonstrated an extension in overall survival rates. Conversely, as far as we know, only one treatment option has been reported to extend overall survival. Instead, they have shown an extension in progression-free survival. We have revised the manuscript to clarify this point more clearly.

Page 20 Line 301

“Despite the similar therapeutic applications of these drugs to existing treatments (49), their aggressive promotion can be attributed to the marginal but clinically significant benefits they offer over competitors, specifically in terms of extension in progression-free survival rather than overall survival rates, a factor crucial in a highly competitive oncology market (50-52).”

References

49. George MA, Qureshi S, Omene C, Toppmeyer DL, Ganesan S. Clinical and Pharmacologic Differences of CDK4/6 Inhibitors in Breast Cancer. Front Oncol. 2021;11:693104.

50. Wang H, Ba J, Kang Y, Gong Z, Liang T, Zhang Y, et al. Recent Progress in CDK4/6 Inhibitors and PROTACs. Molecules. 2023;28(24):8060.

51. Lai H, Jiang W, Zhao J, Dinglin X, Li Y, Li S, et al. Global Trend in Research and Development of CDK4/6 Inhibitors for Clinical Cancer Therapy: A Bibliometric Analysis. J Cancer. 2021;12(12):3539-47.

52. Sledge GW, Jr, Toi M, Neven P, Sohn J, Inoue K, Pivot X, et al. The Effect of Abemaciclib Plus Fulvestrant on Overall Survival in Hormone Receptor–Positive, ERBB2-Negative Breast Cancer That Progressed on Endocrine Therapy—MONARCH 2: A Randomized Clinical Trial. JAMA Oncology. 2020;6(1):116-24.

8) Lines 293-294: Who financed the 2022 guidelines and how many of the people on the committee had COI with one or more makers of drugs used in the treatment of breast cancer?

Reply

Thank you for your valuable comment. The 2022 guidelines were financed by the Japanese Breast Cancer Society, and it has been reported that 115 out of 149 authors (77.2%) on the committee received at least one personal payment from pharmaceutical companies. We have incorporated this point into our revised manuscript.

Page 22 Line 328

“Of note, the guideline was funded by the JBCS itself, and it has been reported that 115 out of 149 authors (77.2%) received at least one personal payment from pharmaceutical companies (57).”

References

57. Murayama A, Higuchi K, Reddy K, Kugo H, Senoo Y. Pharmaceutical company payments to Japanese breast cancer practice guideline2023.

9) Lines 310-320: Direct or indirect gender discrimination should not be tolerated, but the authors frame their argument around the assumption that attending post-clinical evening lectures sponsored by pharmaceutical companies will improve clinical care whereas the content of these talks is often biased in favour of the products made by the company sponsoring the talk.

Reply

Thank you for your feedback. We acknowledge the concerns about the potential bias in content at pharmaceutical-sponsored lectures. Our discussion aims to highlight how these events, despite their biases, significantly influence career advancements and contribute to gender disparities due to their scheduling conflicts. We have now revised our manuscripy as follows:

Page 23 Line 351

“The second author of this paper, who works in the field of breast cancer, has observed how gaining favor with pharmaceutical companies and demonstrating presence at their sponsored lectures play a significant role in career advancement within medical societies in Japan. However, the current scheduling of post-clinical evening lectures significantly hinders the participation of female physicians (59). This scheduling conflict, combined with the dual demands of professional obligations and familial roles, often prevents Japanese women doctors from attending such events, thereby exacerbating gender disparities within the medical community. As a result, unlike other male-dominated surgical fields, although women make up about 40% of breast surgery specialists, men predominantly occupy responsible positions such as directors and presidents within medical societies. To rectify this situation, it is essential to explore more inclusive and accessible approaches to organizing educational events, ensuring equitable participation across genders, and accommodating the diverse needs of the medical community without reliance on pharmaceutical companies.”

Reference

59. Sasaki A. Until I have decided to join the department of breast and thyroid surgery: 

the choice of a female surgeon. Official journal of the Japan Association of Endocrine Surgeons and the Japanese Society of Thyroid Surgery. 2022;39(2):70-5.

Response to reviewer 2

1) Introduction:

- The author should have included a discussion about the potential conflict of interest that may arise from the relationship between the industry and oncologists, and how this can impact patient care. There is a substantial amount of literature on this topic. I understand the authors want to examine the trend of pharma payment, however, I am sure there already a relationship between Japanese breast oncologists and pharmaceutical companies hence mentioning how this can be detrimental to patient care in case not well controlled is important

Reply

Thank you for your valuable comments. We have addressed your concerns in the revised manuscript by including a discussion on the potential conflicts of interest (COIs) that may arise from the relationships between oncologists and the pharmaceutical industry, and how these can impact patient care. We have expanded our introduction to include examples of how COIs have influenced treatment recommendations in Japan, as seen with the continued endorsement of certain treatments despite conflicting international evidence. We believe this addition will enhance the paper's depth and align with the substantial literature on the topic.

Page 9 Line 139

“Such COIs can significantly influence the choice of treatment strategies for breast cancer patients. For example, in t

---

## [Editor Report · Decision Letter 1]

9 Sep 2024

Increasing Trends of Pharmaceutical Payments to Breast Cancer Specialists in Japan: A Retrospective Study from 2016 to 2019

PONE-D-24-07542R1

Dear Dr. Kaneda,

We’re pleased to inform you that your manuscript has been judged scientifically suitable for publication and will be formally accepted for publication once it meets all outstanding technical requirements.

Kind regards,

Narcyz Ghinea

Academic Editor

PLOS ONE

Additional Editor Comments (optional):

I am happy that the authors have adequately addressed the reviewers' feedback.
---

## [Editor Report · Acceptance letter]

17 Sep 2024

PONE-D-24-07542R1 

PLOS ONE

Dear Dr. Kaneda, 

I'm pleased to inform you that your manuscript has been deemed suitable for publication in PLOS ONE. Congratulations! Your manuscript is now being handed over to our production team.

Kind regards, 

on behalf of

Dr. Narcyz Ghinea 

Academic Editor

PLOS ONE